# The Microbiota–Gut–Brain Axis: Gut Microbiota Modulates Conspecific Aggression in Diversely Selected Laying Hens

**DOI:** 10.3390/microorganisms10061081

**Published:** 2022-05-24

**Authors:** Jiaying Hu, Timothy A. Johnson, Huanmin Zhang, Heng-Wei Cheng

**Affiliations:** 1Department of Animal Sciences, Purdue University, West Lafayette, IN 47907, USA; hu165@purdue.edu; 2USDA-Agricultural Research Service, Avian Disease and Oncology Laboratory, East Lansing, MI 48823, USA; huanmin.zhang@usda.gov; 3USDA-Agricultural Research Service, Livestock Behavior Research Unit, West Lafayette, IN 47907, USA

**Keywords:** gut–brain–axis, serotonergic system, gut microbiota, aggression, kynurenine pathway, laying hen, welfare

## Abstract

The gut microbiota plays an important role in regulating brain function, influencing psychological and emotional stability. The correlations between conspecific aggression, gut microbiota, and physiological homeostasis were further studied in inbred laying chicken lines, 6_3_ and 7_2_, which were diversely selected for Marek’s disease, and they also behave differently in aggression. Ten sixty-week-old hens from each line were sampled for blood, brain, and cecal content. Neurotransmitters, cytokines, corticosterone, and heterophil/lymphocyte ratios were determined. Cecal microbiota compositions were determined by bacterial 16s rRNA sequencing, and functional predictions were performed. Our data showed that the central serotonin and tryptophan levels were higher in line 6_3_ compared to line 7_2_ (*p* < 0.05). Plasma corticosterone, heterophil/lymphocyte ratios, and central norepinephrine were lower in line 6_3_ (*p* < 0.05). The level of tumor necrosis factor α tended to be higher in line 6_3_. *Faecalibacterium, Oscillibacter, Butyricicoccus*, and *Bacteriodes* were enriched in line 6_3_ birds, while *Clostridiales* vadin BB60, *Alistipes*, *Mollicutes* RF39 were dominated in line 7_2_. From the predicted bacterial functional genes, the kynurenine pathway was upregulated in line 7_2_. These results suggested a functional linkage of the line differences in serotonergic activity, stress response, innate immunity, and gut microbiota populations.

## 1. Introduction

The gut microbiota plays an important role in the bidirectional interactions between the gut and brain. The microbiota–gut–brain axis involves multiple pivotal pathways, including the autonomic nervous system, the enteric nervous system, the neuroendocrine system, and the immune system [1,2]. The disturbance of the commensal gut microbiome not only causes many intestinal diseases but also impairs brain function, leading to neuropsychiatric disorders with abnormal behaviors [1,3]. Studies conducted in humans and rodents have demonstrated the significant changes in gut microbiota composition between healthy individuals and those with psychological and emotional disorders [4,5]. It has been shown that gut microbiota can affect depressive-like and anxiety-like behaviors directly or indirectly through the regulation of brain functions, such as the activation of the hypothalamic–pituitary–adrenal (HPA) axis, via synthesizing and releasing various neurotransmitters, neuropeptides, neuroactive and immune factors [6,7].

One possible mechanism of gut microbiota influencing psychiatric health is through the tryptophan (TRP) and serotonin (5-HT) metabolism. The function of the central serotonergic system is highly correlated to depression, aggressiveness, impulsivity, and anger-related personality traits in both humans and animals [8,9]. The raphe nuclei are the primary location in the brain for 5-HT production, while the majority of peripheral 5-HT produced by enterochromaffin (EC) cells cannot pass the blood–brain barrier (BBB). Tryptophan passes through the BBB, directly affecting the 5-HT synthesis in the brain, and consequently influencing host behaviors [10,11,12]. Individuals who have diminished levels of 5-HT and 5-hydroxyindoleacetic acid (5HIAA) in the brain are generally associated with increased psychological disorders and related aggression [13,14,15,16]. Previous research has reported that low aggressive mice have higher serotonergic activations compared to high aggressive ones [17,18]. The linkage between 5-HT and psychological disorders has been intensively investigated through the modification of serotonergic regulatory genes and serotonin receptors, as well as the administration of 5-HT reuptake inhibitors [19,20,21]. However, the gut microbiota influencing serotonergic activity and related psychological health has not been thoroughly investigated. In the peripheral system, TRP, as the precursor of 5-HT, can be directly regulated by certain commensal gut microbiota through TRP synthesis, degradation and 5-HT production [22,23]. Commensal bacteria belonging to *Clostridium, Ruminococcus, Blautia*, and *Lactobacillus* have been identified to be able to degrade TRP to tryptamine, a monoamine that is structurally similar to 5-HT [24,25]. Studies on enterochromaffin cell cultures have proved that 1-trytophan hydroxylase (TPH1), which converts peripheral TRP to 5-HT, is induced by short chain fatty acid (SCFA) derived from bacteria [26]. Similarly, the enzyme that converts TRP to N-formylkynurenine, IDO-1, is also induced by commensal gut microbiota [27]. This hypothesis was supported by an acute TRP depletion study conducted in specific pathogen-free (SPF) and germ-free (GF) mice. Compared to GF mice, SPF mice had higher plasma levels of TRP, with less depressive-like behaviors [28].

Inflammation-associated brain dysfunction is another key factor associated with abnormal behaviors, including aggression [29]. A positive relationship between aggression and inflammatory markers, such as interleukin (IL)-6, IL-1ß, IL-2, and tumor necrosis factor (TNF-α), has been reported in both humans and various animals [30,31]. An impaired intestinal barrier increases gut permeability (leaky gut), leading to pathogens and toxic components, such as lipopolysaccharides (LPS), being translocated or diffused systemically, which is one of the reasons for inflammation [32,33]. The leaked bacteria and derived LPS can elevate a series of immune-inflammatory reactions, which are as follows: activating the innate immune system to release pro-inflammatory cytokines and chemokines [34]; and these immune bioactive chemicals can further cross the BBB, causing neuroinflammation; consequently, activating the HPA and SMA (sympathetic-medullary-adrenal) stress response systems [34,35]. Studies in rodents, for example, have shown that GF mice have increased stress responses compared to the mice colonized with a microbial community, which indicates the importance of gut microbiota in regulating stress responses [36,37]. Increased corticotropin releasing hormone (CRH) gene expression and circulating corticosterone (CORT) levels in GF mice are correlated with depression and anxiety-like behaviors [28,38]. Alterations of CORT and catecholamines’ (epinephrine, EP, norepinephrine, NEP, and dopamine, DA) synthesis and release have been identified as the final common pathways in regulating animals’ pathophysiological statue and behavior exhibition [39]. 

Intensive poultry production systems are continuously developed to meet the increasing food demands for an expanding human population [40,41]. However, social stress-induced injurious pecking and cannibalism have become the top health and welfare issues in laying hens reared in all the current housing environments, including both cage and cage-free systems [42,43]. Beak trimming, a routine practice for controlling aggression in the poultry egg industry, has been criticized to cause pain (acute, chronic or both), stress and beak trimming-induced beak malfunction [44]. Understanding the mechanisms of injurious pecking and aggression are critical for developing beak trimming alternatives to improve animal welfare. Various dietary supplements (prebiotics, probiotics, and synbiotics) and fecal microbiota transplantation have been used as bacteriotherapy (or psychomicrobiotic) in humans to target gut microbiota for treating depression, anxiety, autism, and neuropsychiatric disorders [45,46,47]. However, different outcomes have been reported, which could be affected by multiple factors, including host genetics effects on the function of the microbiota–gut–brain axis [48,49]. In chickens, several studies have given arguments about the influence of gut microbiota on modifying host physiology and behaviors. For example, van der Eijk et al. [50] reported that early-life microbiota transplantation influenced feather pecking behaviors, and van Staaveren et al. [51] indicated the potential for *Lactobacillus* spp. as a therapy for feather pecking behaviors. The chicken has been used as a model in developmental biology for over a century for its advantages, such as high productivity, short breeding time, simple management, and a complete genome map [52]. As a result, chickens may very well continue to offer a good animal model for investigating the correlation between genetic variants, gut microbiota population, pathophysiology, and behavioral exhibition associated with psychological and emotional status. 

Numerous studies have found that aggression is heritable [53,54,55]. Differences in injurious behaviors were found in two highly inbred lines of white leghorns selected for resistance (line 6_3_) or susceptibility (line 7_2_) to Marek’s disease (the Avian Disease and Oncology Laboratory, USDA-ARS, East Lansing, MI, USA) [56]. The line differences in production performance and immune functions have also been reported previously [57,58,59]. Compared to line 6_3_ birds, the line 7_2_ birds have a higher number of CD4+ T cells but lower number of CD8+ T cells [60,61]. In addition, the production of cytokine (IL-6 and IL-18) mRNAs in response to Marek’s disease virus infection is different between the two inbred lines, of which the line 7_2_ expresses higher levels of both cytokines than the line 6_3_ [62]. Line differences in social stress-induced aggressive behaviors have also been reported. Compared with chickens from line 7_2_, chickens from line 6_3_ showed fewer aggressive behaviors, including aggressive pecks and fights, as well as duration [15,63]. The difference in behavior expression could be related to line differences in serotonergic activities and the unique distributions of 5-HT1A and 1B receptors [15]. The objectives of the current experiment were to profile the gut microbiota in both of the inbred lines and to evaluate the line’s unique correlations between aggression, gut microbiota, and physiological homeostasis.

## 2. Materials and Methods

### 2.1. Animals and Sample Collection

Two highly inbred lines, 6_3_ and 7_2,_ were used in the current experiment. Chickens from each line were raised in multiple hen colony cages and raised under the same conditions on the USDA, Agricultural Research Service, Avian Disease and Oncology Laboratory (ADOL) specific-pathogen-free farm at East Lansing, Michigan. All these chickens were managed following preapproved standard operating procedures established by the Institutional Animal Care and Use Committee (IACUC, USDA-ARS-ADOL, MI).

Ten sixty-weekk old hens per line were randomly taken for sample collection. A 10-mL blood sample was collected from each chicken through cardiac puncture (n = 10 per group), following sodium pentobarbital sedation (30 mg/kg body weight, i.v.). After blood collection, the chickens were euthanized immediately via cervical dislocation. The raphe nucleus and cecal contents were collected following euthanasia. The blood samples were centrifuged at 700× *g* for 20 min at 4 °C. The brain tissues were immediately put on dry ice, while the cecal content samples were snap frozen in liquid nitrogen. All the samples were stored at −80 °C until later analysis. 

### 2.2. Central Monoamines

The monoamines of NEP, EP, TRP, 5-HT and 5-HIAA of the raphe nuclei were analyzed using high performance liquid chromatography (UltiMateTM 3000 RSLCnano System, Thermo Fisher Scientific Inc., Waltham, MA, USA), following previously described procedures [64]. Briefly, each tissue was weighed and homogenized in ice-cold 0.2 M perchloric acid (PCA) at a 10:1 ratio. The homogenized samples were centrifuged at 18,187× *g* for 15 min at 4 °C. The supernatant of each sample was transferred to a new tube and mixed with the mobile phase (Catalog #: NC0351268, Thermo Fisher Scientific Inc., Waltham, MA, USA) at a 1:1 ratio. The mixture was vortexed for 1 min and centrifuged at 18,187× *g* for 10 min at 4 °C. The supernatant was drawn and filtered through a 0.2 µm polyvinylidene fluoride filter into a sample vial. The mobile phase flow rate was 0.8 mL/min, and the cell potentials were set at 0 mV, 225 mV, and 550 mV. Each sample was tested twice at 10 µL per injection. Each monoamine concentration was calculated according to its corresponding standard and reported as ng/g.

### 2.3. Plasma Serotonin and Tryptophan 

The plasma concentrations of 5-HT and TRP were detected in duplicate using commercially available enzyme-linked immunosorbent assay (ELISA) kits (MyBioSource cat #: MBS016644 and MBS038462, respectively) by following the corresponding protocols. Briefly, 50 µL of standards or samples were added to each well of the antibody pre-coated plate, then 100 µL of horseradish peroxidase-conjugate reagent was added, and the plate was incubated for 60 min at 37 °C. After incubation, the plate was washed and 50 µL of chromogen solutions A and B were added. The plate was placed at dark for 15 min at 37 °C, and then the reaction was stopped by adding a stop solution. The optical density of each sample was read at 450 nm (Epoch Microplate Spectrophotometer, BioTek, Winooski, VT, USA). All the samples were measured in duplicates with a CV ≤ 15% [65]. 

### 2.4. Plasma Cytokines and Immunoglobulin G

The plasma levels of IL-6, IL-2, IL-10, tumor necrosis factor (TNF)- α (Catalog#: MBS037319, MBS005165, MBS007312, MBS9713430, MyBioSource, San Diego, CA, USA), and immunoglobulin G (IgG) (Catalog#: E33-104, Bethyl, Montgomery, TX, USA) were measured using the commercially available ELISA kits following each manufacturer’s protocol. The optical density of each sample was read at 450 nm (Epoch Microplate Spectrophotometer, BioTek, Winooski, VT, USA). All the samples were measured in duplicates with a CV ≤ 15% [65].

### 2.5. Plasma Corticosterone and Heterophil to Lymphocyte Ratio

Plasma corticosterone (CORT) concentration was measured using a commercially available radioimmunoassay kit (Catalog #: 07120103, MP Biomedicals, Solon, OH, USA). The total plasma CORT concentration was analyzed in duplicate following the protocol described by [66]. 

The H/L ratio was determined from 2 blood smears per bird. A thin layer of blood was applied on the surfaces of glass slides and then air dried overnight. The slides were stained with Wright’s staining solution. One-hundred cells, i.e., heterophils and lymphocytes, were differentiated and counted for each slide (total two-hundred cells per bird) using a microscope with 4000× magnification. The H/L ratio was calculated as the number of heterophils/the number of lymphocytes [65].

### 2.6. DNA Extraction and Sequencing

The total bacterial community DNA from frozen cecal contents were extracted using a QIAamp Fast DNA Stool Mini Kit (Qiagen, Hilden, Germany). Briefly, 200 mg of the cecal content sample was homogenized with 1 mL of InhibitEX buffer and 25 uL of proteinase K. The supernatant was mixed with buffer AL and incubated at 70 °C for 10 min. Then, the mixture was purified by adding buffer AW1 and AW2 and eluted in 200 uL of ATE buffer. The total DNA concentration was determined using NanoDrop (ThermoScientific, ND-1000 Spectrophotometer).

Sample 16S rRNA amplicon sequencing was performed by the Environmental Sample Preparation and Sequencing Facility (ESPSF) of Argonne National Laboratory (Chicago, IL, USA). Specifically, the V4 region of the 16S rRNA gene (515F-806R) was PCR amplified with region-specific primers that included sequencer adapter sequences used in the Illumina MiSeq flowcell [67,68]. Briefly, the sample DNA was amplified in a 25 µL PCR reaction system that contains 9.5 µL of MO BIO PCR Water (Certified DNA-Free), 12.5 µL of QuantaBio’s AccuStart II PCR ToughMix (2× concentration, 1× final), 1 µL each Golay barcode tagged Forward Primer and Reverse Primer (5 µM concentration, 200 pM final), and 1 µL of template DNA. The conditions for PCR are as follows: 94 °C for 3 min to denature the DNA, with 35 cycles at 94 °C for 45 s, 50 °C for 60 s, and 72 °C for 90 s; with a final extension of 10 min at 72 °C to ensure complete amplification. The amplicons are then quantified using PicoGreen (Invitrogen) and a plate reader (Infinite^®^ 200 PRO, Tecan). Once quantified, the volumes of each of the products are equally pooled into a single tube. The pool was cleaned up using AMPure XP Beads (Beckman Coulter), and then quantified using a fluorometer (Qubit, Invitrogen). After quantification, the molarity of the pool was determined and diluted down to 2 nM, denatured, and then diluted to a final concentration of 6.75 pM with a 10% PhiX spike for sequencing on the Illumina MiSeq. The amplicons were then sequenced on a 151 bp × 12 bp × 151 bp MiSeq run, using customized sequencing primers and procedures [68].

### 2.7. Data Processing and Analysis

#### 2.7.1. Physiological Data

The physiological data were subjected to one-way ANOVA using the MIXED method of SAS 9.4 software (SAS Institute Inc., Cary, NC, USA), with the genetic line as the major factor. Data transformation (Box-Cox or log transformation) was performed to improve normality [69]. The statistical trend was similar for both the transformed and untransformed data; thus, the untransformed data were reported, and presented as least square means (LSMeans) ± standard error of the mean (SEM). Statistical difference was reported when *p* ≤ 0.05, and a trend was reported when 0.05 < *p* ≤ 0.10. 

#### 2.7.2. 16S rRNA Sequencing Data

The 16S rRNA sequencing data (available at the NCBI SRA database, BioProject accession number PRJNA762159) were analyzed using QIIME2 (version 2020.2) and R studio (version 1.3.1073). The fastq files of the sequenced data were de-multiplexed, quality filtered, and closed reference amplicon sequence variants (ASVs) picking was performed using DADA2, according to the QIIME2 pipeline [70]. In brief, both forward and reverse reads were first trimmed at position (0, 150) and filtered out all the reads with more than 2 expected errors. The filtered reads were then dereplicated and denoised with DADA2 default parameters. After screening and filtering, a total of 497,262 sequences (original sequence was 511,408) remained in the analysis. The table of representative sequences in the samples was produced in QIIME2 and compared with the reference sequence (SILVA database release 132) to produce a feature table containing the frequencies of each taxon per sample [71]. The data were subsampled to 12,804 reads per sample. The measures of alpha diversity (phylogenetic distance, observed ASVs, pilou evenness and Shannon index) were checked for normality and analyzed using ANOVA with a Tukey’s test. A community beta diversity analysis (unweighted Unifrac, weighted Unifrac, Bray–Curtis similarity and Jaccard similarity index) between the samples was performed in R with the “vegan” package [72]. PERMANOVA testing was completed using the adonis function QIIME2 plugin. A multiple test correction q-value was calculated for beta diversity Tukey’s *p*-values and Spearman *p*-values using the Benjamini–Hochberg procedure to control the false discovery rate (FDR). The output matrices were ordinated using principal coordinate analysis (PCoA) in R software. A differential relative abundance analysis of the microbiota composition was carried out with negative binomial regression, using the DESeq2 function within the Phyloseq package (P_adjusted_ < 0.05), where negative and positive log2 fold changes were defined as corresponding groups, respectively [73]. 

PICRUST2 [74] (https://github.com/picrust/picrust2 (accessed on 1 December 2020)) was used to predict the functional profiles based on 16S rRNA sequences between the two genetic lines. The plugin q2-picrust2 was applied in QIIME2 (version 2019.10). MetaCyc pathway abundances and Kyoto Encyclopedia of Genes and Genomes (KEGG) Orthology (KO) were predicted based on the Kyoto KEGG database between the two strains and analyzed with the DESeq2 method (P_adjusted_ < 0.05). All the sequencing data were visualized using the R package “ggplot2” [75].

### 2.8. Correlation Analysis

The relationships between the genus level of the microbiome (relative abundance > 1%), brain neurotransmitters, peripheral 5-HT and TRP, and CORT in each line were calculated for the Spearman’s rank correlation coefficients using the “psych” package in R [76], and the correction was made using Benjamini–Hochberg FDR. For visualization, the “ggplot2” in R was used, where significant (*p* < 0.05) positive and negative correlations are represented in red and blue, respectively.

## 3. Results

### 3.1. Physiological Parameters

#### 3.1.1. Central Monoamines Levels

In the raphe nucleus, line 6_3_-hens tended to have higher levels of EP (*p* = 0.06, Figure 1A), but lower levels of NEP compared to line 7_2_-hens (*p* = 0.002, Figure 1B). Regarding the serotonergic system, line 6_3_-hens had higher levels of 5-HT (*p* = 0.01, Figure 1C) but no difference in 5HIAA (*p* = 0.16, Figure 1D), resulting in a lower 5HIAA/5-HT conversion rate (*p* = 0.04, Figure 1E) in 6_3_-hens. Compared to line 7_2_-hens, the line 6_3_-hens also tended to have a higher level of TRP than line 7_2_-hens (*p* = 0.08, Figure 1F).

#### 3.1.2. Peripheral Serotonergic, Stress and Immune Parameters

In the blood samples, TRP was higher in line 6_3_-hens than in line 7_2_-hens (*p* = 0.03, Figure 2A). No difference in plasma 5-HT was found between the two lines (Table 1). The plasma CORT level was lower in line 6_3_-hens than line 7_2_-hens (*p* = 0.04, Table 1). The heterophil to lymphocyte ratio was also lower in line 6_3_ than line 7_2_ (*p* < 0.0001, Table 1). The hens from the line 6_3_ tended to have higher TNF-α than those from line 7_2_ (*p* = 0.09). No difference was found in IL-6, IL-2, IL-10, and IgG (*p* > 0.05) between the lines (Table 2).

### 3.2. Microbiota Profile

#### 3.2.1. Alpha Diversity

The Faith phylogenetic distance was lower in line 6_3_-hens than line 7_2_-hens (*p* = 0.03, Figure 2A). The observed ASV, Shannon index and Pielou evenness did not differ between the line 6_3_ and line 7_2_ (*p* > 0.05, Figure 2B–D). 

#### 3.2.2. Beta Diversity

Regarding the beta-diversity between the two genetic lines, with the *p*-value adjusted using the Benjamini–Hochberg FDR correction approach, there were significant differences in Jaccard (*p* = 0.001, q = 0.001, Figure 3A), Bray–Curtis (*p* = 0.001, q = 0.001, Figure 3B), weighted UniFrac (*p* = 0.002, q = 0.004, Figure 3C) and unweighted UniFrac (*p* = 0.001, q = 0.002, Figure 3D) distances.

#### 3.2.3. Taxonomic Composition

At the phylum level, the majority of 16S rRNA amplicons belonged to five main phyla, including the following: *Bacteroidetes*, *Cyanobacteria, Firmicutes, Proteobacteria,* and *Tenericutes* (Figure 4A). *Bacteroidetes* and *Firmicutes* were two most abundant phyla in both lines.

At the genus level, a deeper analysis using DESeq2 identified 119 out of 993 ASVs that significantly differed between the line 6_3_ and line 7_2_ hens (P_adjusted_ < 0.05, Figure 5). With a log_2_ fold-change larger than 2, *Faecalibacterium, (Eubacterium) hallii group, Oscillibacter, Butyricicoccus, Anaeroplasma, Bacteriodes, Parasutterella, Peptococcus, Ruminiclostridium 5, Ruminococcaceae, Ruminococcaceae UCG-005, Ruminococcaceae UCG-008, Ruminococcaceae UCG-009 and Tyzzerella* were more enriched in line 6_3_, while *Alistipes, Anaerofilum, Anaerostipes, Clostridiales vadinBB60 group, Clostridium sensu stricto 1, Dielma, Family XIII AD3011 group, Lachnoclostridium, Lachnospiraceae FCS020 group, Methanomassiliicoccus, Mollicutes RF39, Oscillospira, Romboutsia, Ruminococcaceae UCG-010, Ruminococcus 1 and Ruminococcus 2* were more enriched in line 7_2_.

More ASVs from the genus *Fournierella* (two out of three ASVs)*, Lachnospiraceae* (nine out of thirteen ASVs)*, Ruminococcaceae* UCG-014 (four out of seven ASVs), were enriched in line 6_3_, while more ASVs from the genus *Lactobacillus* (three out of four ASVs), *and Christensenellaceae* R-7 group (two out of three ASVs) were more enriched in line 7_2_. 

#### 3.2.4. Metagenome Functions Prediction

Predicted functional metagenomic profiles based on MetaCyc pathways were generated using PICRUSt2. The comparisons between the lines 6_3_ and 7_2_ resulted in the identification of 57 metabolic pathways that differed between the lines of hens (*p* < 0.05, log_2_ fold change > 1 shown in Figure 6). The predicted metagenomic function pathways, including antibiotic resistance, generation of precursor metabolites and energy, carbocylate degradation, inorganic nutrient metabolism and biosynthesis pathways (coenzyme A, amino acid, aromatic compound, cofactor, carrier and vitamin, fatty acid and lipid, and secondary metabolite) were increased in line 72, while CO2 fixation and biosynthesis pathways (carbohydrate, and nucleoside and nucleotide) were increased in line 63.

Specifically, the enzymes related to the TRP metabolism pathway varied between the two lines. In line 7_2_ hens, tryptophan 2, 3-dioxygenase (K00453: TDO2, log_2_ fold change = −2.35, *p* < 0.0001), tryptophanase (K01667: tnaA, log_2_ fold change = −0.41, *p* = 0.003), indolepyruvate decarboxylase (K04103: ipdC, log_2_ fold change = −1.59, *p* = 0.02) and monoamine oxidase (K00274: MAO, log_2_ fold change = −3.10, *p* = 0.01) were detected more often, while aldehyde dehydrogenase (K00128: ALDH, log_2_ fold change = 0.27, *p* = 0.01) was increased in line 6_3_ hens (Figure 7).

#### 3.2.5. Microbial and Neurotransmitter Correlations

The correlations within and between the relative abundance of individual bacterial genera and concentrations of physiological relevant molecules (Figure 8) show that brain 5-HT (r = −0.53, *p* = 0.02) and EP (r = −0.66, *p* = 0.002) were negatively correlated with the *Clostridiales vadin* BB60 group, which was the most prominent bacteria in line 7_2_; and brain NEP was negatively correlated to *Faecalibacterium* (r = −0.69, *p* = 0.001), *Fournierella* (r = −0.78, *p* < 0.0001), but positively correlated to *Romboutsia* (r = 0.76, *p* = 0.003).

## 4. Discussion

The gut microbiome plays an important role in regulating physiological and behavioral homeostasis [1]. The data in this study showed that the gut microbiota (microbiome) community composition and diversity are correlated with the differences in synthesis of central and peripheral TRP and 5-HT, cytokines, and stress indicators between the two genetic lines of hens diversely selected for or against Marek’s disease, or viral-induced tumor formation. These differences are also correlated with each line’s unique characteristics of productivity, disease resistance, and aggression reported previously [14,15,54,56,57,58].

The implications of serotonergic functions in behaviors have been the subject of many studies in both humans and animals in the last century. The outcomes from multiple experimental methods, including neurosurgical lesion of the raphe nucleus, and pharmacological or genetic inhibition of 5-HT synthesis, reuptake, and binding capability of 5-HT receptors, indicate that the hypofunction of the central serotonergic system is associated with increased aggressive and impulsive behaviors [77,78]. Previous studies in our laboratory have demonstrated some line differences in serotonergic activity and aggressive behaviors between the line 6_3_ and 7_2_ chickens. The chickens from line 7_2_ exhibited increased aggressive behaviors with lower brain 5-HT concentrations, compared to line 6_3_ chickens [15,54]. In the current study, we also found that hens from the line 7_2_ (high aggressive) had lower levels of 5-HT and TRP in the raphe nuclei than that of the line 6_3_ hens (low aggressive). A similar genetic variant in brain monoamines was also reported between highly aggressive and non-aggressive mice, in which excessively aggressive mice had low levels of 5-HT in the prefrontal cortex [17]. A similar heritable pattern of TRP was also found in the peripheral system, where hens from line 6_3_ had higher plasma TRP than line 7_2_. Tryptophan, the major precursor of 5-HT, can pass the BBB and be directly involved in brain 5-HT production [5,79,80]. The tryptophan-serotonin deficiency hypothesis of aggression has been evidenced in patients with depression [81,82] and a naturalistic 5-HT deficient animal model, the tryptophan hydroxylase 2 knockin mouse [83]. Unlike TRP, no difference was found in circulating 5-HT between the line 6_3_ and line 7_2_ hens. Because 5-HT cannot cross the BBB, whether the level of peripheral 5-HT contributes to aggression is unknown. In addition, the pathophysiological roles of 5-HT in behavioral and motivational regulation are different between the central and peripheral serotonergic systems [13,84]. Contradicting trends of blood 5-HT concentrations have been reported in association with behavioral dysfunctions, including aggressiveness. Studies comparing aggressive and non-aggressive dogs showed lower serum concentrations of 5-HT in aggressive dogs [85]. However, increased circulating 5-HT was found in an aggressive strain in laying hens [86] and in human patients with severe psychological illness and related aggression [87]. 

The difference in TRP and 5-HT synthesis and metabolism between the two genetic lines could be attributable to the commensal microbiota variance. Approximately 90% of 5-HT is produced by the gut enterochromaffin cells (EC), and gut microbiota plays a crucial role in TRP synthesis, degradation and 5-HT production, which consequently impact peripheral and central 5-HT concentrations [88,89,90]. At the genus level, many of the bacterial genera were different between the line 6_3_ and line 7_2_ hens. *Faecalibacterium*, the most prominent genera that was enriched in line 6_3_, has been associated with the behavior of the host. Patients suffering from bipolar depression and major depressive disorder (MDD) have reduced levels of *Faecalibacterium* [91,92]. In a clinical study with multiple cohorts, the fecal level of *Faecalibacterium prausnitzii* (the sole species of *Faecalibacterium* genus) was negatively correlated with the severity of depression between depressive and healthy subjects [93]. More ASVs were enriched in the genus *Lachnospiraceae*, another beneficial microbiota, which was also found in line 6_3_ birds, and its function is closely related to 5-HT and TRP metabolism. A mouse model of autism with deficient TRP and 5-HT metabolism in the intestines has defined a significant decrease in *Lachnospiraceae* taxa [94]. Other studies conducted in depression, bipolar and anhedonia susceptible patients, who generally have decreased serotonergic activities, reported decreased *Lachnospiraceae* abundance compared with the healthy controls [95,96,97]. Merging data have suggested the potential interventions of *Lachnospiraceae* and *Faecalibacterium* genera for the treatment of psychological disorders, including anxiety, depression, and autism [98,99]. Other enriched bacteria in the line 6_3_, *Eubacterium hallii* group was also found to have a lower abundance in psychological disorder patients than healthy controls [100,101,102]. Microbial metabolites, including SCFA and long chain fatty acids (LCFA), are known to affect psychological health. Some of the genera, such as *Faecalibacterium* and *Butyricicoccus*, are butyrate-producing organisms. SCFA producers can directly stimulate 5-HT secretion [50,103,104]. In the current study, central 5-HT, and both central and peripheral TRP, were found to be higher in the line 6_3_ hens, together with the fact that line 6_3_ birds were less aggressive than line 7_2_ [15]; this could partly be due to the higher relative abundance of the beneficial bacteria composition in the gut of line 6_3_ birds.

In line 7_2_, the more dominant genera, *Anaerostipes*, *Clostridium* and *Lachnoclostridium* are consistently found to be higher in MDD patients [105]. The genus *Romboutsia* is highly involved in metabolic capabilities, including carbohydrate utilization, single amino acids fermentation, and metabolic end products [106]. The enrichment of this bacteria could directly lead to the heavier body weight and early sexual maturation found in line 7_2_ birds. However, another dominated genus *Clostridiales vadin* BB60 group has been poorly classified for its biological effectiveness [107,108]. To our knowledge, the *Clostridiales vadin* BB60 group is only defined to be associated with an increased risk of developing alopecia universalis, an autoimmune disorder [108].

Elevated stress hormones likely also contribute to the lower serotonergic activity in the 7_2_ birds. In the current study, we observed that the line 7_2_ hens had a higher plasma CORT and H/L ratio compared to 6_3_ hens, the two common stress indicators used in avian species, which suggested an elevated stress status in the line 7_2_ hens. The difference in the stress responding pattern is also indicated by the different changes in neurotransmitters, where the NEP level in the raphe nuclei was significantly higher in the line 7_2_ hens, compared to the line 6_3_. The activation of the HPA stimulates the synthesis of CRH and further regulates central NEP and peripheral CORT [109,110]. Both NEP and CORT, as stress hormones, could contribute to the behavior variance between the two genetic lines. In human and rodent studies, the hypersecretion of NEP and cortisol/corticosterone are found in most stress-related psychiatric disorders with suppressed central 5-HT activity [111,112]. Moreover, elevated stress hormones can also lead to disrupted gut barrier function and altered commensal bacteria [113]. An increase in proinflammatory bacteria, such as *Helicobacter* and *Streptococcus*, and a decrease in anti-inflammatory bacteria, such as *Roseburia* and *Lachnospiraceae* species, are observed in individuals under stress conditions [114]. In the current experiment, the levels of NEP had a positive correlation with the *Clostridiales VadinBB60* group (enriched in line 7_2_) but a negative correlation with *Faecalibacterium* and *Fournierella* (both enriched in line 6_3_).

The immune system plays an important role in both brain and the gut integrity and is involved in regulating the function of the microbiota–gut–brain axis via the immune-brain connections [115]. The gut microbiota and related metabolites are the key regulators by which the GIT communicates with the immune system [116,117]. Recent studies have suggested that the role for the gut microbiota in the regulation of inflammation is to modulate the differentiation of various types of both immune and inflammatory cells for cytokine production and hematopoiesis [118,119,120]. For example, polysaccharide A (PSA) derived from *Bacteroides fragilis* directly influences the immune system, leading to systemic T cell deficiencies and Th1/Th2 imbalances in lymphoid tissues [121]. A *Lactobacillus cripatus*-based probiotic enhances the anti-inflammatory response of both macrophages and epithelial cells by releasing anti-inflammatory cytokine, IL-10, and reducing proinflammatory cytokines, TNF-α and IL-6 [122]. *Faecalibacterium prausnitzii* administration increases the IL-10 level and reduces IL-6 as well as CORT levels in depression-like and anxiety-like rats [123]. However, the other enriched genera in line 6_3_ *Anaeroplasma* and *Peptococcus* were opportunistic pathogens, which elicited various host immune responses in human diseases [124,125]. In the current study, a trend of increased proinflammatory cytokine, TNF-α, was determined in the less aggressive 6_3_-hens compared to the 7_2_-hens. On the other side, the more abundant bacteria strains in the line 7_2_ could have an important immunomodulation role in the intestinal mucosa, such as some beneficial *Lactobacillus* species [126]. For example, *Lactobacillus rhamnosus* and *Lactobacillus gasseri* have shown promising anti-inflammatory abilities by inhibiting proinflammatory cytokines [127,128,129]. Thus, the proinflammatory cytokine might be lower due to the presence of these beneficial *Lactobacillus* strains. More research is needed to further identify the correlation between cytokine production and gut microbiota in the two genetic lines.

PICRUSt2 prediction indicates the different metabolic pathways involved in energy and nutrition metabolism between the line 6_3_ and line 7_2_ chickens. The majority of the pathways related to biosynthesis and energy metabolism were increased in relative abundance in line 7_2_, which could contribute to the growth and production differences between the two genetic lines. Importantly, the pathway of TRP metabolism varied between the two lines. In line 7_2_, the predicted abundance of tryptophanase (tnaA) and indolepyruvate decarboxylase (ipdC) genes increased, by which TRP can be catalyzed into indole, pyruvate, and or ammonium [130]. In addition, the abundance of 2, 3-dioxygenase (TDO2), the rate-limiting enzyme to metabolize TRP to kynurenine, also had a log_2_ fold increase of over 2 in the line 7_2_ birds. These results indicate that there could be a higher degradation rate of TRP to kynurenines through the activated kynurenine pathway, rather than 5-HT synthesis [131]. The dysregulation of TRP metabolism, with shifts in the balance between the kynurenine and 5-HT pathways, is associated with psychiatric and neurological disorders [132,133]. The deletion of TDO2 in mice resulted in lower anxiety-like behaviors than the wild type [130]. Imbeault et al. [134] also reported reduced anxiety and cognitive deficits after the inhibition of TDO2 expression. The higher TDO2 gene abundance predicted in line 7_2_ might be a possible factor that contributes to the higher aggression reported previously [15]. Additionally, corticosteroid is known to enhance the gene expression of the enzyme TDO2 [135], which agrees with the higher CORT level found in the line 7_2_ birds in the current study. Moreover, monoamine oxidase (MAO) also increased in abundance in line 7_2_; this could be one of the reasons that they have lower central and peripheral 5-HT levels. However, these hypotheses need to be tested in future studies. 

The current results discovered that the ubiquity of host-associated microorganisms indicate that microbes play an important role in chicken behavior, and chickens’ aggression could be reduced by modifying their microbial composition via the microbiota–gut–brain axis. However, the most behavioral differences found among chickens are affected by multiple factors, including the rearing systems. The different housing systems with inherently different designs cause chickens to exhibit different behaviors [43,136]. This approach deserves further attention under the current commercial conditions, i.e., chickens kept in both the enriched cage system and the cage-free systems (various aviaries), with the aim that eventually management practices, such as beak trimming, will no longer be necessary.

In conclusion, this work has identified the unique correlations of gut bacteria and related metabolic pathways with the different pattens of stress response, neurotransmitter expression, and innate immunity between the two diversely selected lines of chickens. Compared with the aggressive line 7_2_, the line 6_3_ has inherently higher central tryptophan-serotonergic activities, lower stress hormones and increased certain major beneficial bacteria, such as *Faecalibacterium* and *Lachnospiraceae*. In addition, the metagenomic function predicted certain gut microbes that could be related to the tryptophan-serotonergic metabolic differences between the two lines. The findings of this study have provided new signals for developing novel biotherapy strategies for controlling aggression in laying hens by identifying gut bacteria and their metabolites correlated with behavioral repertoire.

## Figures and Tables

**Figure 1 microorganisms-10-01081-f001:**
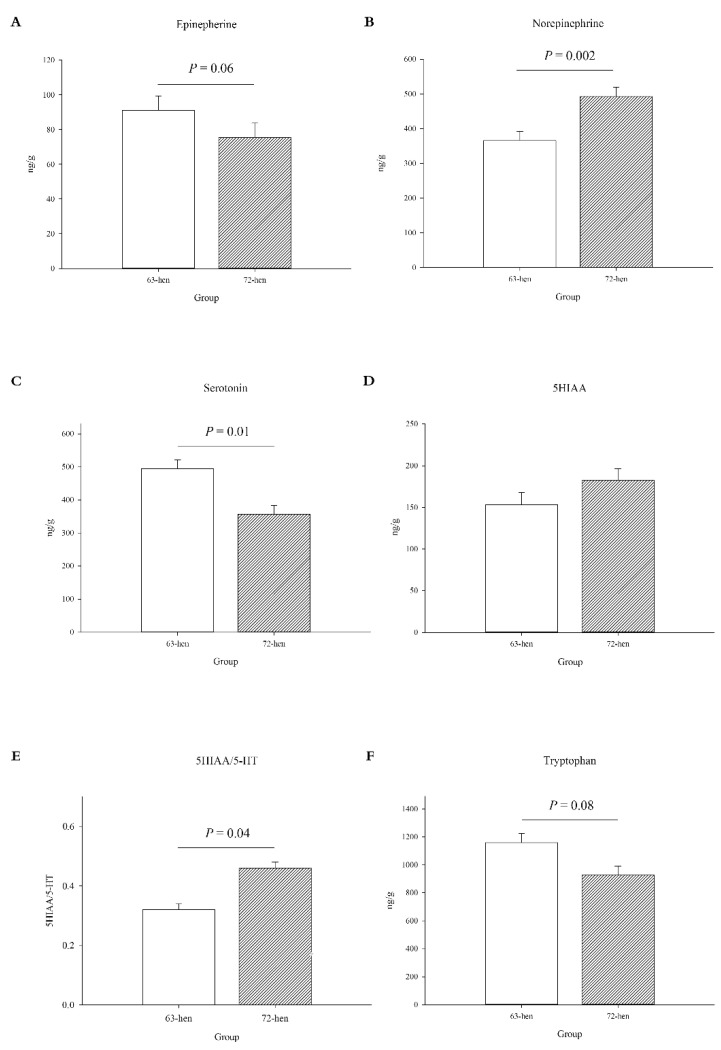
The measured neurotransmitters and neuropeptides in the raphe nucleus of two diversely selected chicken lines. (**A**), epinephrine (EP), (**B**), norepinephrine (NEP), (**C**), serotonin (5-HT), (**D**), 5-hydroxyindoleacetic acid (5HIAA), (**E**), 5HIAA/5-HT ratio, (**F**), tryptophan (TRP). Data are presented as: LSM + SEM (n = 10).

**Figure 2 microorganisms-10-01081-f002:**
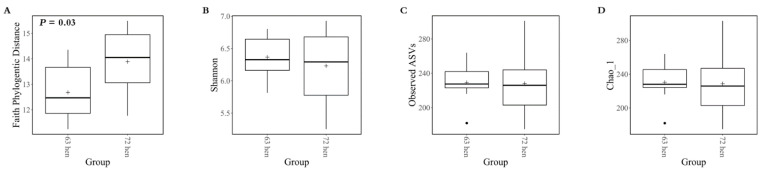
Alpha diversity of cecal content bacteria in two diversely selected chicken lines. Box-and-whisker plots of diversity within samples. (**A**): estimated number of amplicon sequence variants (ASVs) calculated by the Chao1 index, (**B**): alpha diversity calculated by the Shannon index, (**C**): species evenness calculated by the Pielou’s evenness measures; and (**D**): phylogenetic diversity calculated by Faith’s phylogenetic diversity (PD).

**Figure 3 microorganisms-10-01081-f003:**
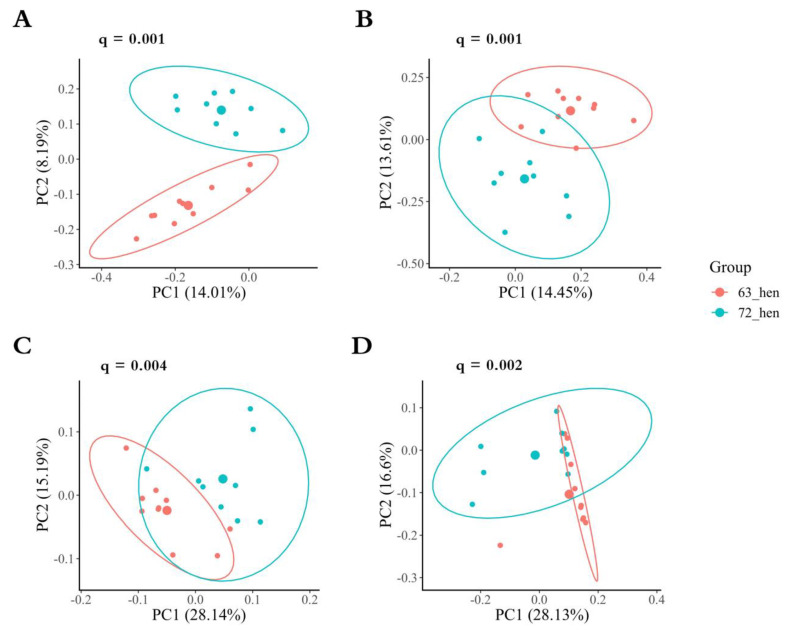
Principal coordinates analysis (PCoA), which shows the similarity between two diversely selected chicken lines. (**A**): Bray–Curtis similarity, (**B**): Jaccard similarity, (**C**): unweighted UniFrac, and (**D**): weighted UniFrac distances with 95% confidence ellipses.

**Figure 4 microorganisms-10-01081-f004:**
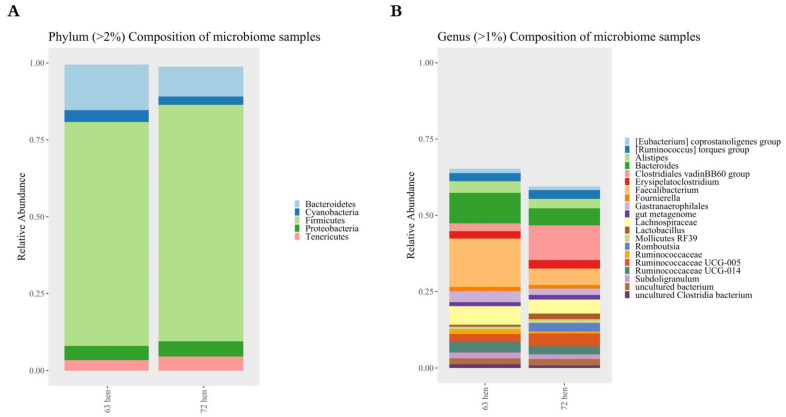
The different cecal bacterial compositions between two diversely selected chicken lines (n = 10). (**A**) phyla, and (**B**) genera analyses that compose an average of >2% and >1%, respectively, of the community, while other minor phyla or genera are not shown.

**Figure 5 microorganisms-10-01081-f005:**
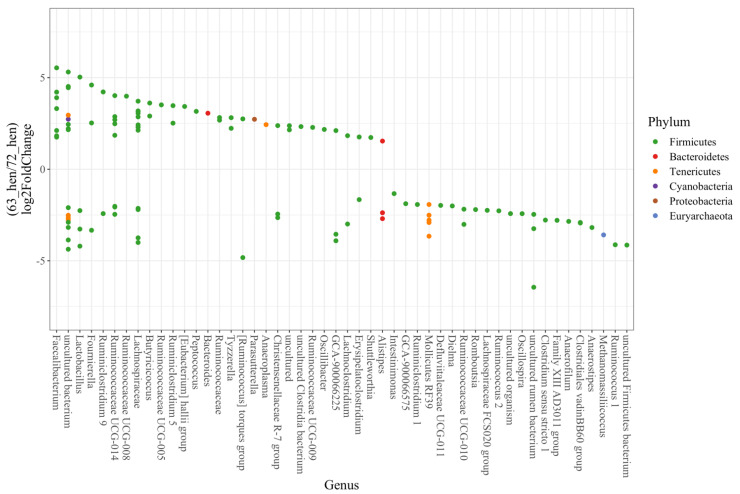
Pairwise comparisons (DESeq2 analysis) between two diversely selected chicken lines (n = 10). Differential abundant OTUs (*p* < 0.05) in 6_3_-hen and 7_2_-hen are shown. OTUs were assigned to genus (x-axis) and phylum level (colors). Positive “log_2_ Fold Change” values (y-axis) indicate higher abundance in 6_3_-hens, and negative values indicate higher abundance in 7_2_-hens.

**Figure 6 microorganisms-10-01081-f006:**
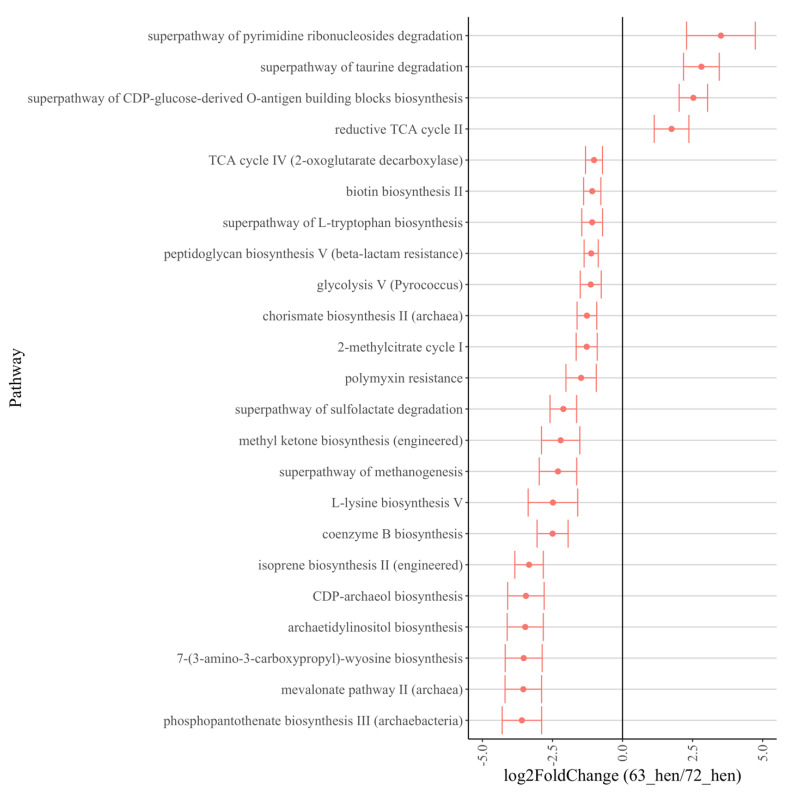
MetaCyc pathway difference predictions using 16S rRNA data with PICRUSt2 between hens from line 63 and line 72 (>1 log2 fold change). Positive “log2 fold change” values (x-axis) indicate higher expression of certain pathways in 63-hens, and negative values indicate higher expression in 72-hens.

**Figure 7 microorganisms-10-01081-f007:**
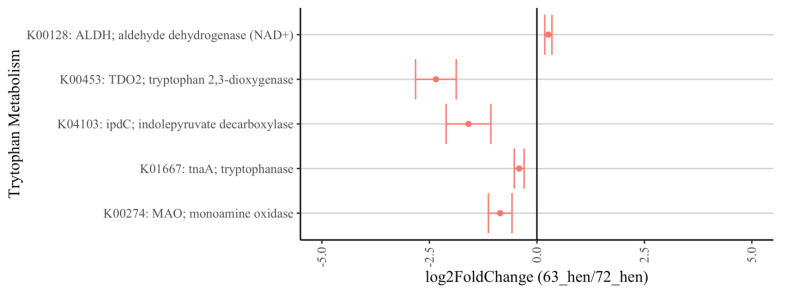
Relative abundance of tryptophan metabolism on KEGG orthology between diversely selected 6_3_-hens and 7_2_-hens. Positive “log_2_ fold change” values (x-axis) indicate higher expression of certain pathways in 6_3_-hens and negative values indicate higher expression in 7_2_-hens.

**Figure 8 microorganisms-10-01081-f008:**
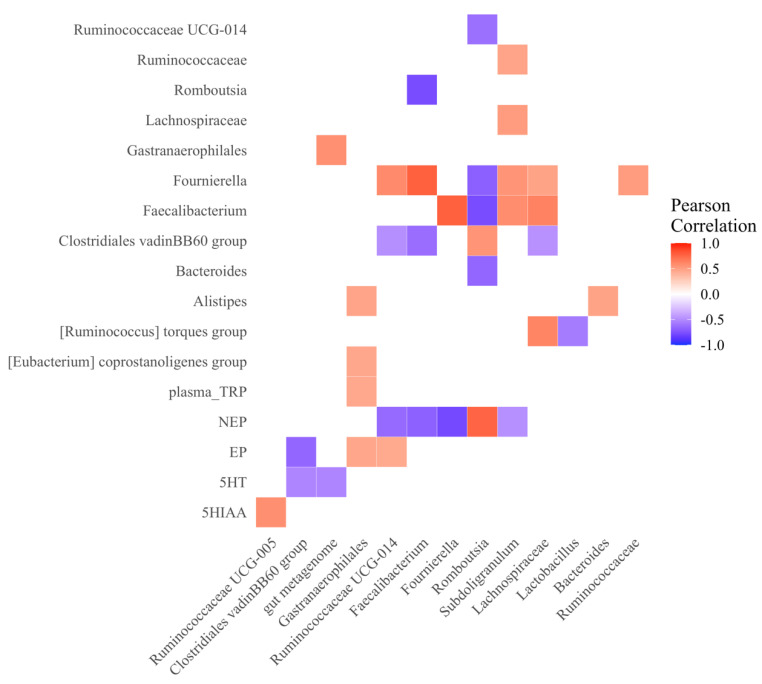
Heatmap showing the Spearman’s rank correlation coefficients between genera abundance and physiological parameters. Blue and red colors indicate significant negative and positive correlations, respectively (*p* < 0.05).

**Table 1 microorganisms-10-01081-t001:** Peripheral serotonin, tryptophan, corticosterone and heterophil/lymphocyte ratio levels between two diversely selected chicken lines.

f	5-HT (ng/mL) ^1^	TRP (µmol/mL) ^1^	CORT (ng/mL) ^1^	H/L Ratio ^1^
6_3_-hen	61.38	171.52 ^a^	8.44 ^b^	0.16 ^b^
7_2_-hen	59.46	121.42 ^b^	9.75 ^a^	0.50 ^a^
SEM	3.79	15.37	1.51	0.04
n ^2^	10	10	10	10
*p*-Value	0.73	0.03	0.05	<0.0001

^a,b^ Least squares means within a column for the 2 strains lacking a common superscript differ (*p* < 0.05). ^1^ 5-HT: serotonin, TRP: tryptophan, CORT: corticosterone, H/L ratio: heterophil to lymphocyte ratio. ^2^ Average number of observations per least squares mean.

**Table 2 microorganisms-10-01081-t002:** Peripheral immunoglobulin G and cytokine levels between two diversely selected chicken lines.

	IgG (mg/mL) ^1^	IL-6 (pg/mL) ^1^	IL-2 (pg/mL) ^1^	IL-10 (pg/mL) ^1^	TNF-α (ng/mL) ^1^
6_3_-hen	12.0	28.14	60.09	9.37	36.65
7_2_-hen	12.9	27.56	71.65	13.13	30.73
SEM	0.73	1.63	12.8	1.64	2.37
n ^2^	10	10	10	10	10
*p*-value	0.54	0.81	0.54	0.12	0.09

^1^ IgG: immunoglobulin G, IL: interleukin, TNF-α: tumor necrosis factor alpha. ^2^ Average number of observations per least squares mean.

## Data Availability

The 16S rRNA sequencing data are available at the NCBI SRA database, BioProject accession number PRJNA762159.

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
