# Peer review of "The Microbiota–Gut–Brain Axis: Gut Microbiota Modulates Conspecific Aggression in Diversely Selected Laying Hens"

_microorganisms, 2022, doi:10.3390/microorganisms10061081_

Round 1

Reviewer 1 Report

An interesting study examining the links between animal behaviour, the gut microbiota and metabolites. 

Some of the English and words used needs amending and several paragraphs in the Introduction and Discussion are a long list of other study's findings. Use the other studies to tell the story and support your experimental design and results. 

Replace "mental" with "psychological" throughout the manuscript.

Abstract

"prediction" to predictions; and the second to last sentence has "predicted" twice.

Introduction

First paragraph is good.

The second and third paragraphs do not tell the story of the study and are a long list of other study observations.

The terminology usually used is "blood-brain barrier" not "brain-blood"

"has been further evidenced in" is incorrect grammar, try "was supported by"

"As one of the reasons," is incomplete - reasons for what? The rest of the sentence needs re-structuring as I do not understand the message.

"microbiota-gut-brin" to "microbiota-gut-brain"

What do you mean by "simple management"?

Tables 1 and 2 have no headings

Discussion

"were increased in presence" change to "were detected more often"

Page 15 - two very long paragraphs that need dividing and improving the connection with the other study results and the present study. They read like a list.

Remove "Indeed," as it does not make sense

Was there more than one study in the Rosado 2010 reference?

"In a cohort study" does not make sense and does not connect with the 3 references? Describe the cohort study so the reader can understand why you are citing it.

Delete "lay an explicit effect" and replace with "affect"

Change "dominated" to dominant

What do you mean by "poorly classified" for biological effectiveness?

"compared to 63 hens" is in the wrong place, move to after "ratio".

Throughout the discussion be clear on the design and methodology of the references when you cite them.

I do not understand the sentence beginning "On the other side, "

Author Response

Reviewer 1:

An interesting study examining the links between animal behaviour, the gut microbiota and metabolites. 

Some of the English and words used needs amending and several paragraphs in the Introduction and Discussion are a long list of other study's findings. Use the other studies to tell the story and support your experimental design and results. 

Replace "mental" with "psychological" throughout the manuscript.

Done as suggested.

Abstract

"prediction" to predictions; and the second to last sentence has "predicted" twice.

Done as suggested.

Introduction

First paragraph is good.

The second and third paragraphs do not tell the story of the study and are a long list of other study observations.

Thank you for the comments. The second paragraph, we would like to make a clear background introduction how psychological health is related to the serotonergic system and immunology, which is the key information of our hypotheses.

The terminology usually used is "blood-brain barrier" not "brain-blood"

Done as suggested.

"has been further evidenced in" is incorrect grammar, try "was supported by"

Done as suggested.

"As one of the reasons," is incomplete - reasons for what? The rest of the sentence needs re-structuring as I do not understand the message.

Sentence has been rephrased.  

"microbiota-gut-brin" to "microbiota-gut-brain"

Done as suggested.

What do you mean by "simple management"?

Simple management here means there is no additional management strategies added, such as environmental enrichment and feed additives.

Tables 1 and 2 have no headings

Thank you for the comments. Headings have been added.

Discussion

"were increased in presence" change to "were detected more often"

Done as suggested.

Page 15 - two very long paragraphs that need dividing and improving the connection with the other study results and the present study. They read like a list.

Thanks for the comments. We have revised the two paragraphs.

Remove "Indeed," as it does not make sense

Done as suggested.

Was there more than one study in the Rosado 2010 reference?

To best of our knowledge, we only found Rosado et al 2010 study in dogs, however, there are studies in other species with similar hypotheses.

"In a cohort study" does not make sense and does not connect with the 3 references? Describe the cohort study so the reader can understand why you are citing it.

Thank you for the comments. We have rephrased the sentence to be more clear.

Delete "lay an explicit effect" and replace with "affect"

Done as suggested.

Change "dominated" to dominant

Done as suggested.

What do you mean by "poorly classified" for biological effectiveness?

“poorly classified” here indicates that the biological effectiveness of clostridiales vadin BB60, unlike other genera, have not been comprehensively studied.

"compared to 63 hens" is in the wrong place, move to after "ratio".

Done as suggested.

Throughout the discussion be clear on the design and methodology of the references when you cite them.

Thank you for the comment. We have checked the references.

I do not understand the sentence beginning "On the other side, "

This sentence means there are several abundant genera that could have an immunomodulation role in the intestinal mucosa, by which it may cause the difference between the two lines.

Reviewer 2 Report

Dear Editor(s), the manuscript hereby submitted discloses a very interesting and present topic. The fact that the gut brain axis is sometimes underrated in regards to animals enhances the importance of these type of studies. Also, by decreasing the aggressiveness in animals also diminishes their stress and well being, which ultimately influences the final products. In addition to know in advance which types have a higher trend for such behaviours or disorders and how to prevent/treat them, especially avoiding drugs is also of high importance.

The manuscript is well written and structure, few very minor errors detected and therefore, I recommend its publications

Author Response

Thanks for the comments and encouragement.

Reviewer 3 Report

The article is well written and highlights an interesting aspect of Gut-brain axis.
The concern is these are more circumstantial evidence. Therefore the discussion needs to address how in spite of being in an in vivo situation where the factors in the pathways are regulated by several other aspects how does the particular strain emphasize its beneficial effect. The positive effects maybe because of other factors as well.

Limitations of the study as well needs to be highlighted. 

Author Response

Thanks for the encourage, and we agree with the comments, there could be some limitations of the study (or strains of chickens) for the commercial egg production. It has been addressed as the followings.

“The current results discovered the ubiquity of host-associated microorganisms indicate that microbes play an important role in chicken behavior, and chickens’ aggression could be reduced by modified their microbial composition via the microbiota-gut-brain axis.  However, the most behavioral differences found among chickens are affected by multiple factors, including the rearing systems. The different housing systems with inherently designs affecting chickens to exhibit their behavior differently (Janczak and Riber, 2015; Campbell et al., 2019). This approach deserves further attention under the current commercial conditions, i.e., chickens kept in both the enriched cage system and the cage-free systems (various aviaries), with the aim that eventually management practices such as beak trimming will no longer be necessary.”

Campbell, D. L. M., E. N. de Haas, and C. Lee. 2019. A review of environmental enrichment for laying hens during rearing in relation to their behavioral and physiological development. Poult. Sci. 98: 9-28.

Janczak A. M. and Riber A. B. 2015. Review of rearing-related factors affecting the welfare of laying hens. Poult. Sci. 94: 1454-1469.